# Effect of *Salmonella* Typhimurium Colonization on Microbiota Maturation and Blood Leukocyte Populations in Broiler Chickens

**DOI:** 10.3390/ani12202867

**Published:** 2022-10-20

**Authors:** Kelsy Robinson, Anna L. F. V. Assumpcao, Komala Arsi, Gisela F. Erf, Annie Donoghue, Palmy R. R. Jesudhasan

**Affiliations:** 1Poultry Research Unit, ARS, USDA, Mississippi State, MS 39762, USA; 2Department of Poultry Science, University of Arkansas, Fayetteville, AR 72701, USA; 3Poultry Production and Product Safety Research Unit, ARS, USDA, 1260 W Maple St, Fayetteville, AR 72701, USA

**Keywords:** *Salmonella*, *Salmonella* Typhimurium, microbiome, lymphocytes, broiler, chicken

## Abstract

**Simple Summary:**

Reducing *Salmonella* levels in chickens is vital to reducing the number of human *Salmonella* infections resulting from contact with contaminated chickens and poultry products. The intestinal tract of chickens is home to a diverse population of bacteria that serve as one of the first lines of defense against pathogenic microbes. Therefore, we sought to understand changes in the intestinal bacterial populations following *Salmonella* infection. Our results showed a clear change in the intestinal bacterial structure after *Salmonella* colonization. An inverse relationship between *Salmonella* and *Lactobacillus* and *Escherichia* was observed as well as an increase in *Bacteroides.* Additionally, we provide insight into the timing of the host immune response with monocytes and thrombocytes acting early following colonization followed by heterophils. In total, this data demonstrates the ability of *Salmonella* to change intestinal microbiota composition and highlights genera that may be useful as probiotics to fight *Salmonella* colonization.

**Abstract:**

Reducing *Salmonella* in commercial chickens is vital to decreasing human salmonellosis infections resulting from contact with contaminated poultry and poultry products. As the intestinal microbiota plays an important role in preventing pathogen colonization, we sought to understand the relationship between *Salmonella* infection and the cecal microbiota and the host immune system. Day-of-hatch broiler chicks were assigned to three treatments: control, artificial (SA), and natural (SN) *Salmonella* infection. At seven days of age, control and SA birds were inoculated with PBS or *Salmonella* Typhimurium, respectively. Five SA birds were transferred to SN cages to facilitate natural infection. Cecal content and blood samples were collected at 0, 8, 14, and 21 days of age for microbiota and leukocyte analysis, respectively. A significant change in microbiota composition was observed in both groups as noted by a decrease in *Lactobacillus* and *Escherichia* and an increase in *Bacteroides*. Leukocyte analysis revealed a decrease in the percentage of circulating monocytes at 7 days post-infection while a decrease in thrombocyte and an increase in heterophil percentages were seen at 14 days post-infection. Taken together, these results demonstrate the ability of *Salmonella* to modulate the intestinal microbiota to facilitate colonization. Additionally, results indicated an early role of monocytes and thrombocytes during colonization, followed by heterophils.

## 1. Introduction

Non-typhoidal *Salmonella* is a Gram-negative bacterium commonly transmitted to humans through direct contact with animals and contaminated food such as raw meat, poultry, raw eggs, raw milk, and other dairy products [1]. *Salmonella* has more than 2500 serovars, and most of the serovar-causing human foodborne illnesses fall under the subspecies, *enterica* (I) [2]. In the United States, the Centers for Disease Control found that 50 out of 186 *Salmonella* outbreaks from 2017 to 2020 were due to contact with backyard poultry resulting in 4411 illnesses and 870 hospitalizations outbreaks [3]. Most *Salmonella* serovars in the chicken gastrointestinal tract are asymptomatic to chickens but cause diarrheal disease in humans [4]. According to USDA-ERS (2021), the total estimated cost of human foodborne illness due to non-typhoidal *Salmonella* in 2018 was ~4.1 billion dollars [5]. The current approaches to reducing the non-typhoidal serovars in poultry have had limited success. The target of the US Department of Health and Human Services—Healthy People 2020 was to reduce the incidence of *Salmonella* from 15 per 100,000 cases to 11.1 per 100,000 cases by 2020, but the incidence increased to 17.1 in 2019 [6,7]. The non-typhoidal serovars remain a major foodborne pathogen globally. Despite strides to prevent *Salmonella* in poultry production, outbreaks continue to occur, necessitating the development of novel intervention tools for controlling them in chickens.

The chicken’s gastrointestinal tract comprises complex and diverse bacterial populations, which provide many beneficial functions to the host, including conferring colonization resistance against invading pathogenic microorganisms [8]. The first microbes in the chick gut are thought to be vertically transmitted from the hen through the yolk sac or absorbed through the shell from the environment [9]. Next-generation sequencing of embryos has revealed the presence of Gram-positive *Micrococcus*, *Bacillus*, and *Enterococcus* during the end stages of incubation [10], while *Enterococcus* [11] and members of the gram-negative family *Enterobacteriaceae* have been identified in the intestinal tract of newly hatched chicks [12,13,14]. However, the current practice of placing commercial eggs in a relatively sterile incubator and hatcher limits the number and type of bacteria capable of gaining access pre-hatch. This presents the chicks with an empty ecological niche increasing the opportunity for early colonization by pathogens [15]. Without exposure to the hen or nest environments, microbes from the eggshell and hatching and rearing environments serve as the first primary source of microbiota for colonization after hatching [15,16]. Following the stabilization of microbiota composition, the resident microbes prevent colonization by pathogenic microbes through mechanisms such as competitive exclusion and habitat filtering [17]. However, factors such as antibiotic supplementation [18,19,20], stress [21], and high pathogenic insult [17] can disrupt these mechanisms allowing pathogen colonization to occur.

While multiple studies have been undertaken to understand the relationship between the *Salmonella* serovar Enterica and the chicken intestinal microbiota [8,22,23,24], only a few have included *Salmonella* Typhimurium (ST) [25,26,27]. Furthermore, these studies have focused primarily on dietary interventions to reduce ST infection and are lacking meaningful investigations into the interaction of ST with the microbiota [25,26,27]. Therefore, we sought to fully characterize the microbiota’s role in ST-challenged and naturally colonized birds. We also determined the effect of *Salmonella* challenge on the immune status of broiler chickens to evaluate the birds’ natural methods to control *Salmonella* and prevent infection in humans.

## 2. Materials and Methods

### 2.1. Bacteria and Culture Conditions

Three nalidixic acid (NA) resistant field strains of ST were used for this study (ATCC 14028 NA resistant, ATCC 14028 WT NA resistant, and an NA-resistant field strain) (ATCC, Manassas, VA, USA). Bacterial strains were inoculated separately in tubes containing tryptic soy broth and incubated overnight at 37 °C. Overnight cultures were mixed in equal portions, centrifuged to remove spent media, and resuspended in phosphate-buffered saline at a concentration of ~1 × 10^7^ CFU/mL for chicken oral gavage.

### 2.2. Animals and Experimental Design

The live bird experiments and procedures were approved by the University of Arkansas’ institutional animal care and use committee under the approval number Ag-IACUC #21029. 

To determine changes in the cecal microbiota and health status of broilers during *Salmonella* colonization, 120 1-day-of-hatch Cobb by-product breeder chicks (females) were obtained from a local hatchery and raised for 21 days in state-of-the-art environmental and temperature-controlled USDA colony houses. Chicks were vaccinated in ovo for Marek’s disease. Birds were provided floor space that equals or exceeds FASS guide recommendations at each stage of life (e.g., 135 in^2^ at 40–43 days of age). Chicks were placed on fresh pine shavings and kept in an environmentally controlled room and provided with age and species-appropriate conditions in accordance with the Cobb Broiler Management guide [28]. All birds were provided food and water ad libitum throughout the study.

Upon placement, birds were divided into three groups: (1) negative control (NC; 45 birds), (2) *Salmonella* challenge (SA; 35 birds), and (3) *Salmonella* natural colonization (SN; 35 birds). At 7 days of age, birds in the negative control and *Salmonella* challenge groups were orally gavaged with 0.25 mL of phosphate buffered saline or the 3-strain mixture of ~1 × 10^7^ CFU/mL ST, respectively. Following ST inoculation, five chicks from the SA group were moved to SN pens to act as seeder birds. Blood and cecal content samples were collected from the NC group on days 0 and 6 and from all groups on days 8, 14, and 21. Samples were collected from five birds per treatment at each time point, and samples were used for the determination of immune response, enumeration of *Salmonella*, and cecal microbiota evaluation.

### 2.3. Sample Collection

For sample collection, birds were euthanized by inhalation of carbon dioxide. Blood samples were collected for serum and plasma preparation in appropriate tubes, while EDTA-blood samples were used for flow cytometry analysis. Cecal contents were collected for microbiological and microbiota analysis. For microbiological analysis, cecal contents were plated on XLD agar plates to enumerate *Salmonella*. For microbiota analysis, cecal contents were collected in a DNA/RNA shield (Zymo Research, Irvine, CA, USA). DNA was isolated using the ZymoBIOMICS DNA Miniprep Kit (Zymo Research, Irvine, CA, USA) according to the manufacturer’s protocol. DNA quality and quantity were determined using Nanodrop. Two DNA samples from day 0 were found to be of low quality and were not used for sequencing leaving three samples for the day 0 timepoint.

### 2.4. Flow Cytometry Analysis

All antibodies were purchased from SouthernBiotech (SouthernBiotech, Birmingham, AL, USA) unless otherwise specified. Samples were prepared as described previously by Seliger, et al. [29]. Briefly, 20 μL of EDTA-blood was diluted in 980 μL of staining buffer (PBS pH 7.2, 1% bovine serum albumin, 0.1% NaN_3_). Diluted blood was mixed 1:1 with appropriate antibody mixtures (1:100) and incubated for 45 min at 4 °C protected from light. Four combinations of mouse-anti-chicken antibodies were used to identify (1) thrombocytes and leukocytes: SPRD-CD45 (LT40) and FITC-CD41/61 (11C3, Bio-Rad, Hercules, CA, USA), (2) T cells: PE-γδTCR (TCR1), FITC-CD4 (CT-4), and SPRD-D8α (CT-8), (3) αβ T-cells and γδ T-cells: PE-γδTCR (TCR1), FITC-αβTCR (αβ1 (TCR2) and αβ2 (TCR3)), and SPRD-CD45 (LT40), (4) B cells, monocytes, and heterophils: SPRD-CD45 (LT40), FITC-KUL01 (KUL01), and PE-Bu1 (AV20). Heterophils were identified based on forward and side scatter characteristics of CD45+ cells. After incubation, samples were further diluted in 150 μL of staining buffer and kept on ice protected from light until measurement. Samples were acquired using a BD C6-plus-Accuri flow cytometer (Beckton Dickinson, San Jose, CA, USA), and results were analyzed using FlowJo v10.8.1 software.

### 2.5. Microbiota Analysis

Purified DNA samples were shipped on dry ice to the core facility at the University of Texas Southwestern Medical Center (Dallas, TX, USA) for sequencing of the V3–V4 region of the 16S rRNA gene using the PE300 primer set (341f: CCTACGGGDGGCWGCAG, 806r: GACTACNVGGGTMTCTAATCC). Samples were prepped using the Zymo Quick-16s Plus NGS Library Prep Kit (Zymo Research, Irvine, CA, USA) and sequenced on an Illumina MiSeq machine. Following demultiplexing, raw reads were imported in QIIME2 [30] and processed using the deblur algorithm [31]. The deblur denoise-16S option was used with sequences truncated to 403 nucleotides based on quality scores. Denoised sequences referred to as amplicon sequence variants (ASVs) were classified using the SILVA 138 database [32].

Analysis and visualization of microbiota composition were accomplished using R version 4.2.1 [33]. All plots were made using ggplot2 version 3.3.6 [34]. For α-diversity analysis, data were normalized using cumulative sum scaling (CSS) in the metagenomeSeq package of R [35] and calculated using Observed ASVs and Shannon index in the phyloseq package 1.4.0 [36]. For β-diversity and differential abundance analysis, data were normalized using the centered log ratio (CLR) according to the methods of Gloor, et al. [37]. Principle component analysis (PCA) was used to visualize differences in microbiota composition.

### 2.6. Statistical Analysis

Statistical significance was measured using parametric and nonparametric methods depending on data normality and equality of group variances. Significant differences in α-diversity were determined using the Kruskal-Wallis test, while permutational multivariate ANOVA (permanova) in the vegan package of R [38] was used for β-diversity analysis. Significant differences in abundance between groups were determined using the ANOVA-like differential expression tool ALDEx2 [39]. For flow analysis, all statistical analyses were conducted using GraphPad Prism v9.4.1. The Student’s *t*-test was used to determine the statistical significance between negative control and *Salmonella*-challenged groups. *p*-values ≤ 0.05 were considered statistically significant.

## 3. Results

### 3.1. Variation in Cecal Microbial Populations and Leukocyte Sub-Populations with Age

Following sequencing, a total of 7,781,641 sequences were obtained, with an average of 123,518 sequences per sample (range of 95,149 to 188,969). To confirm the consistency of our data with what is known regarding the chicken intestinal microbiota, we first investigated microbial composition changes with age.

#### 3.1.1. Effect of Age on Cecal Microbiota Diversity

To account for the compositionality of microbiota data [37], sequence counts were normalized using the CLR method, and differences in microbiota composition were visualized using a PCA plot (Figure 1A). A clear separation was observed between D0 and all other samples indicating the uniqueness of the hatch microbiota. Beginning on day 6, microbiota composition was shown to change gradually, with day 6 more similar to day 8 than day 21. Statistical evaluation using permanova revealed changes to be significant (*p* < 0.05). However, the R^2^ value was low (0.203), which is most likely an indication of the small sample size used in this study.

Changes in species richness and evenness with age were measured using the Observed Features (Figure 1B) and Shannon diversity (Figure 1C) calculators, respectively. Both indices increased with age, beginning with a sharp significant (*p* < 0.05) increase from day 0 to day 6. Interestingly, species richness further increased from day 6 to day 8 (*p* < 0.05) before beginning to plateau on days 14 and 21. A slight, but non-significant increase in evenness was observed between days 6 and 8 before reaching a plateau.

#### 3.1.2. Shifts in Phylum and Genus Composition with Age

Bacterial composition was investigated by calculating the relative abundance of CSS normalized data at the phylum (Figure 2A) and genus (Figure 2B) levels. A notable switch in composition was observed at both levels. At the phylum level, the cecal microbiota at day-of-hatch was dominated by Proteobacteria in two of the three birds investigated (two samples were not sequenced due to low-quality DNA). Proteobacter was replaced by Firmicutes beginning on day 6 with the proportion of Firmicutes ranging from 55 to 99%. For days 8 through 21, Firmicutes were observed to dominate all samples representing greater than 86% of all sequences.

In agreement with the alpha diversity analysis, only two genera were detected on day 0. *Escherichia* was the primary genus detected in two of the three samples, while the third was predominantly *Enterococcus* with lower levels of *Escherichia*. The abundance of both genera decreased as cecal diversity increased with age. Beginning on day 6, members of the genera *Oscillospira,* unclassified *Clostridiales*, and unclassified *Ruminococcaceae* became the dominant cecal members, collectively representing over 50% of the microbiota. This composition remained relatively stable through day 21 with only slight variations in specific genera observed.

#### 3.1.3. Alteration in Percentages among Leukocyte Populations with Age

To evaluate immune development with age, the percentage of various leukocyte subpopulations was investigated using flow cytometry of whole blood from 10 control birds per timepoint (Figure 3). Results revealed a significant shift in percentages of leukocyte populations between days 0 and 6, which then stabilized for the remainder of the study. Specifically, the percentage of heterophils decreased significantly (*p* < 0.0001) from greater than 75% on day 0 to less than 40% on day 6. This was associated with a significant (*p* < 0.001) increase in CD4^+^ T cells which remained significantly different from day 0 through day 21. A numerical increase in the percentage of monocytes began on day 6 and became significantly different from day 0 (*p* < 0.001) on days 14 and 21. The stability of leukocyte populations was confirmed by the lack of significant difference in cell percentages between other combinations of days, except for an increase in the percentage of monocytes between days 8 and 14. Additionally, no significant differences were observed for the B cell, CD8^+^ T cell, and λδ T cell percentages.

### 3.2. Effect of Salmonella Colonization on Microbiota and Leukocyte Populations

#### 3.2.1. Microbial Diversity and Composition

To determine the interaction between the role of the microbiota in *Salmonella* colonization, birds were inoculated with a three-strain mixture of ST at 7 days of age via oral gavage. Following inoculation, five birds were transferred to a pen containing naïve birds to allow for horizontal transmission. Birds receiving the oral gavage were considered to be *Salmonella* challenged and are referred to as the SA group. Birds colonized via horizontal transmission were considered to be naturally colonized and are referred to as the SN group. Colonization was confirmed by plating cecal contents at 21 days of age which revealed a cecal *Salmonella* load of 3.89 and 3.78 log CFU/mL for the SA and SN groups, respectively. For microbiota analysis, birds were euthanized, and cecal contents were collected on days 8 (24 hpi), 14 (7 dpi), and 21 (14 dpi) of life. PCA analysis revealed a clear separation in microbial composition between control and *Salmonella*-infected birds at all three timepoints (Figure 4A–C). The separation was observed to be statistically significant (*p* = 0.001) at all three timepoints with R^2^ values of 0.276, 0.276, and 0.262 for days 8, 14, and 21, respectively. Alpha diversity analysis revealed a significant decrease in species richness on day 8 (*p* < 0.05) with only approximately 80 ASVs detected in the SA group compared to approximately 160 in the control birds (Figure 4D). ASV numbers began to recover by 7 dpi with only numerical decreases in ASVs detected in the *Salmonella* groups on days 14 and 21. The Shannon index did not reveal any significant differences in species evenness among treatments at any time point (Figure 4F).

Differences in microbiota composition were further investigated by determining changes in the relative abundance of bacterial genera (Figure 4F). The average relative abundance of the top 20 genera at each time point is also described in Appendix A. A two-phase change in the cecal microbiota was observed following artificial *Salmonella* inoculation. An initial disruption was observed at 24 hpi (day 8) and was marked by an increase in *Lactobacillus*, *Bacteroides*, and unclassified *Bacillaceae* in SA birds versus control. This was associated with a decrease in the genera *Oscillospira*, *Subdoligranulum*, *Enterococcus*, and *Escherichia*. At 7 dpi (14 days of age), a second shift in microbiota composition was observed in which the bloom of *Lactobacillus* and unclassified *Bacillaceae* was lost with the abundance of both classes being notably lower in SA birds than that of control. Concomitantly, a relative abundance of *Subdoligranulum*, unclassified *Lachnospiraceae*, and *Butyricoccus* in SA birds were observed to be greater than that of the control. Increases in *Subdoligranulum,* unclassified *Lachnospiraceae,* and *Butyricoccus* were still present at 21 days of age, while *Lactobacillus* and unclassified *Bacillaceae* returned to levels similar to that of control. Interestingly, the effect of SA on *Oscillospira*, *Bacteroides,* and *Escherichia* abundance remained consistent across time points.

Natural *Salmonella* infection via vertical transmission resulted in a shift in microbiota composition similar to that of phase two described above. Similar to the SA group, a decrease in the relative abundance of *Lactobacillus*, *Escherichia,* and unclassified *Bacillaceae* and a strong increase in *Bacteroides* were observed at 14 and 21 days of age, while no change in *Oscillospiraceae* or *Subdoligranulum* was observed at either time point. A temporary increase in members of the *Butyricoccus* and *Enterococcus* genera and a decrease in unclassified *Lachnospiraceae* occurred at 14 days of age. However, the abundance of *Butyricoccus* was similar to the control by 21 days of age, while *Enterococcus* and unclassified *Lachnospiraceae* levels switched, becoming less than and greater than controls, respectively.

Significant changes in specific bacterial taxa were determined using ALDEx2 on CLR-transformed data at the ASV level. Genera with significantly affected ASVs (*p* < 0.05 and effect <|1|) in at least one comparison are shown in Figure 5. In agreement with the relative abundance data above, the largest number of significantly affected ASVs was seen when comparing SA and control birds at 8 days of age with 28 ASVs belonging to 13 different genera significantly increased or decreased. The number of significantly different abundant ASVs in the SA group decreased to 13 and 4 at days 14 and 21 of age, respectively. In contrast to the SA group, the number of significantly affected ASVs in SN birds versus control increased from 14 (2 ASVs) to 21 days of age (7 ASVs). Interestingly, most of the ASVs in the SN group were also significantly different from SA birds at both time points, indicating a unique microbiota response to natural colonization.

#### 3.2.2. Time-Dependent Response of Leukocyte Subpopulations to Salmonella Inoculation

Host immune response to *Salmonella* inoculation was evaluated using flow cytometry of stained whole blood. Several samples were found to have begun the clotting process during transfer to the lab and were excluded from the analysis. Unequal sample numbers were taken into account during analysis and results revealed time-dependent changes in leukocyte populations. No significant difference in total leukocytes, thrombocytes, or various leukocyte populations was observed on day 8 (data not shown). By day 14 (7 days post-inoculation, Figure 6A), a significant decrease in the percentage of monocytes was observed (*p* < 0.05), while all other populations remained the same. Monocyte percentage levels then increased with no significant difference observed between control and SA on day 21 (14 days post-inoculation, Figure 6B). However, a significant decrease (*p* < 0.05) in the percentage of thrombocytes and an increase in the percentage of heterophils (*p* < 0.05) were observed on day 21.

## 4. Discussion

Human salmonellosis, caused by a non-typhoidal *Salmonella* infection, is responsible for approximately 26,500 hospitalizations and 420 deaths each year [40]. Of these infections, *Salmonella* serovar Typhimurium is one of the most prevalent causative agents [41]. As contact with poultry and contaminated poultry products is a common source of human *Salmonella* infections [1], controlling *Salmonella* colonization is important to decrease human illness. The intestinal microbiota confers multiple benefits to the host, including facilitation of nutrient breakdown and uptake, strengthening of the intestinal barrier, conferring colonization resistance against pathogenic bacteria, and modulation of immunity [42,43]. Therefore, we sought to understand the role of the intestinal microbiota during ST colonization as well as its effect on host immunity.

Numerous studies have been conducted to understand chicken intestinal microbiota and its relationship with health and disease. Microbiota composition is affected by numerous factors such as sex [44], genetic line [45], diet [46], environment [43], and host stress [47,48], yet age is consistently the primary driving factor with the microbiota undergoing successional changes during the life cycle of a bird [12,43,49]. Interestingly, only a few studies have investigated the bacterial populations of day-of-hatch chicks [12,13,14]. These studies have revealed the initial cecal microbiota to be dominated by Gram-negative Proteobacteria, primarily *Enterobacteriaceae* [12,13,14]. As the microbiota rapidly develop and become more diverse within the first week of life, Proteobacteria is replaced by Firmicutes, with Clostridia dominating the rest of life [14]. In concordance with this, successional changes in cecal microbiota diversity were observed in this study. Proteobacteria members *Escherichia* and *Enterococcus* were the only genera detected in day-of-hatch birds, and their abundance decreased significantly with age as Firmicutes members and overall diversity increased.

Both ST challenge and natural colonization of the chicken intestinal tract with ST affected microbiota composition. Oral gavage with *Salmonella* resulted in a two-phase shift in microbiota composition, which included a sharp initial disruption followed by a second, more stable change. The stress associated with handling young poultry causes an increase in cortisol, [50] which can directly affect microbiota composition [51]. Additionally, the sudden introduction of large numbers of pathogenic bacteria may overwhelm colonization resistance mechanisms within the established microbiota resulting in compositional changes [17]. Interestingly, microbial profiles of the naturally colonized group closely resembled that of the second phase in the artificial group, indicating that the dynamics of microbiota changes during infection are different with artificial being more abrupt. However, the similarities in the final composition indicate that the initial stress from oral gavage does not affect the long-term composition. Contrary to our results, Liu and others [52] did not find a dramatic shift in composition at 1 dpi *Salmonella* Enteritidis oral inoculation. This difference is most likely due to the difference in bacterial serovar (Typhimurium vs. Enteritidis) or chicken breed (Cobb vs. Jining Bairi chickens) used in each study.

Previous studies investigating *Salmonella* Enteritidis have found significant changes in microbiota composition following colonization [8,22,53]. These have included increases in *Enterobacteriales* [8,22,53] and decreases in the *Clostridiales, Lactobacillales,* and *Bifidobacteriales* orders [53] and the *Enterococcaceae, Lachnospiraceae, Peptostreptococcaceae, Ruminococcaceae,* and *Erysipelotrichaceae* families [8,22]. When comparing the mechanics of *Salmonella* low-, intermediate, and super-shedders, Kempf and colleagues [23] found a sharp decrease in fecal *Enterococcus* sp. following infection and considered *Enterococcus* abundance pre-infection to be a biomarker of *Salmonella* shedding. Consistent with this, we found a decrease in *Enterococcus* as well as *Lactobacillus, Escherichia,* and *Bacillaceae* following artificial and natural ST colonization. An increase in *Lactobacillus* was also observed in the ileum of ST-infected birds fed a highly nutritious diet when compared to a poor nutritious diet [54]. Interestingly, Zhang et al. [27] found ST colonization in the ileum to have no effect on *Lactobacillus* and to decrease *Escherichia* abundance versus control. Investigation of the effect of ST on the fecal microbiota of laying hens revealed an increase in genera such as *Subdoligranulum, Shuttleworthia, Sellimonas,* Ruminiclostridium_9, *Intestinimonas, Faecalibacterium, Enorma,* and *Blautia* [55]. A concomitant increase in *Oscillibacter, Flavonifractor, Erysipelatoclostridium, Eisenbergiella, Caproiciproducens,* and *Butyricoccus* was also observed [55].

Only the genus *Bacteroides* was found to be increased by ST in this study. Previous work has found propionate produced by *Bacteroides* to be involved in the microbiota’s colonization resistance to ST in mice [56]. Therefore, the increase in *Bacteroides* in this study may indicate the microbiota’s attempt to prevent lasting ST colonization. An inverse relationship between *Salmonella* and *E. coli* was also observed by Litvak and colleagues [57] who revealed the ability of *E. coli*, in the presence of butyrate-producing *Clostridia,* to prevent *Salmonella* infection by competing for oxygen. While no changes in *Clostridia* were observed in this study, the decrease in *E. coli* may still reflect competition for oxygen resources following *Salmonella* infection.

Finally, the investigation into the effect of ST colonization on host immune response revealed a significant decrease in the percentage of peripheral blood monocytes at 7 dpi. At 14 dpi, monocyte levels had recovered, and a significant decrease in the percentage of thrombocytes and an increase in the percentage of heterophils were observed. Monocytes, thrombocytes, and heterophils are all part of the innate immune system with the ability to kill microbes through phagocytosis, release of granule contents, and production of inflammatory cytokines [58]. An increase in blood monocytes following oral *Salmonella* infection has previously been seen in mice [59], while heterophils have been found to be an important factor in controlling *Salmonella* infections in poultry [60] via the release of the antimicrobial peptide Cathelicidin-2 [61]. A decrease in thrombocytes following *Salmonella* infection has been observed previously in humans and mice and may reflect the consumption of thrombocytes as they form aggregates around pathogenic cells [62].

## 5. Conclusions

Control of *Salmonella* colonization in commercial poultry is vital to reducing human cases of salmonellosis. The microbiota plays a key role in preventing bacterial infections through competitive exclusion and modulation of the host immune response. Here we found ST challenge and natural colonization to produce similar effects on the cecal microbiota of broiler chickens. Changes were characterized by a decrease in *Enterococcus, Lactobacillus, Escherichia,* and *Bacillaceae* and a concomitant increase in the propionate-producing genus *Bacteroides.* Additionally, we showed chicken blood leukocytes to have a two-phase response to ST infection with the percentage of monocytes, heterophils, and thrombocytes significantly affected. Taken together, these data provide further insight into the role of the cecal microbiota and host immunity in ST colonization.

## Figures and Tables

**Figure 1 animals-12-02867-f001:**
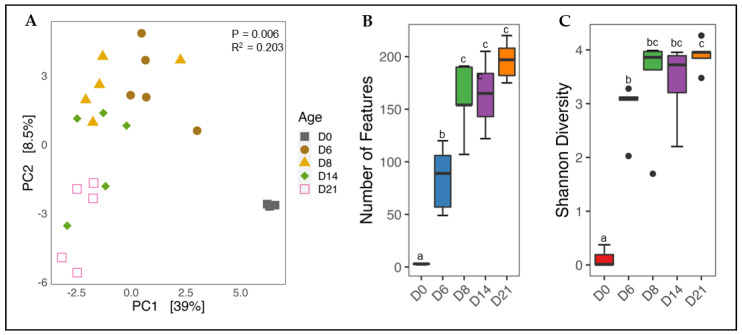
Temporal changes in cecal microbiota diversity. Cecal samples from non—infected chicks were used to determine variation in microbiota composition with age. For beta diversity, the distance between samples was calculated using the Euclidean distance of CLR normalized data and visualized with a PCA plot (**A**). Community richness and evenness were calculated using the Observed Features (**B**) and Shannon diversity (**C**) calculators, respectively. Significant differences in beta diversity were determined using permanova. A Kruskal-Wallis test was used to evaluate alpha diversity, and means were separated with pairwise Wilcoxon Rank-Sum tests. Segments not sharing a common superscript are considered significantly different (*p* < 0.05). Samples falling outside of the first and third quartiles are shown as black dots.

**Figure 2 animals-12-02867-f002:**
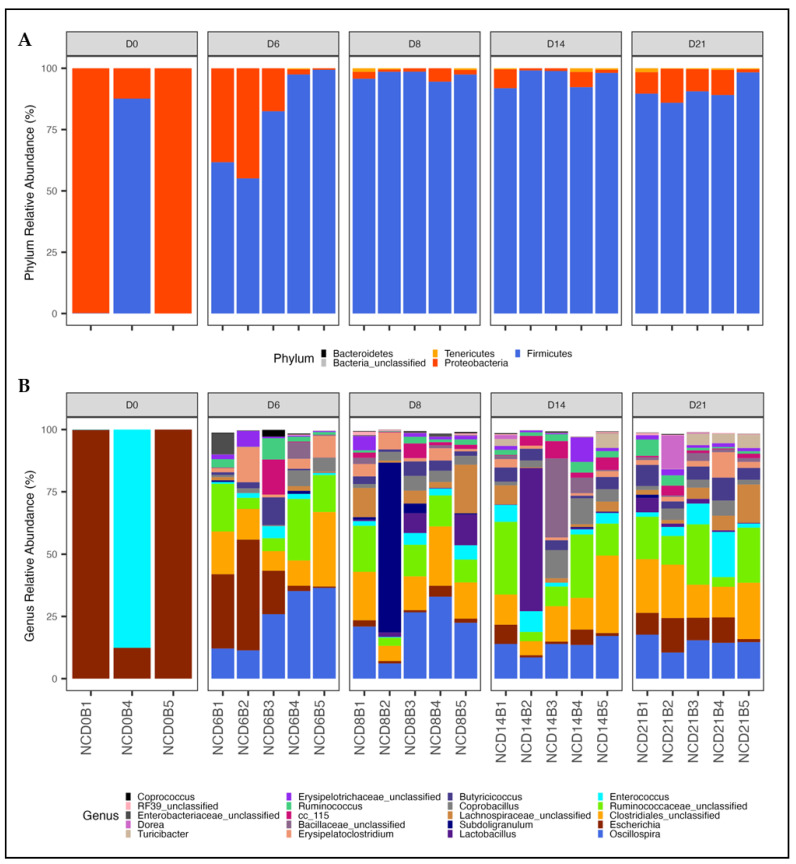
Variation in bacterial composition with age. Bacterial composition was visualized by calculating the relative abundance of CSS normalized data at the phylum (**A**) and genus (**B**) levels using 3 to 5 birds per age. Only the top 20 genera are shown.

**Figure 3 animals-12-02867-f003:**
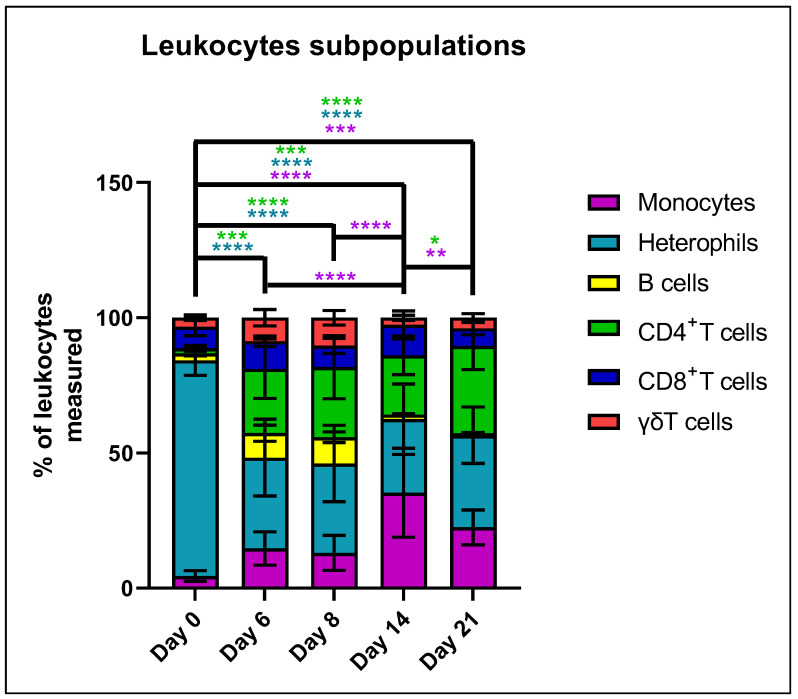
Characterization of temporal shifts in the proportions of chicken leukocyte populations. Whole blood, collected from 10 control birds at each time point, was subjected to flow cytometry to identify changes in leukocyte subpopulations. Results are shown as the percent of total leukocytes measured (CD45+CD41/61−). Significant differences were determined using Two-way ANOVA, and pairwise comparisons were completed using Tukey HSD. Results were considered significantly different at a *p* < 0.05. Significant difference for each cell type are indicated by asterisks color-coded to match the cell type (* = *p* < 0.05, ** = *p* < 0.01, *** = *p* < 0.001, **** = *p* < 0.0001).

**Figure 4 animals-12-02867-f004:**
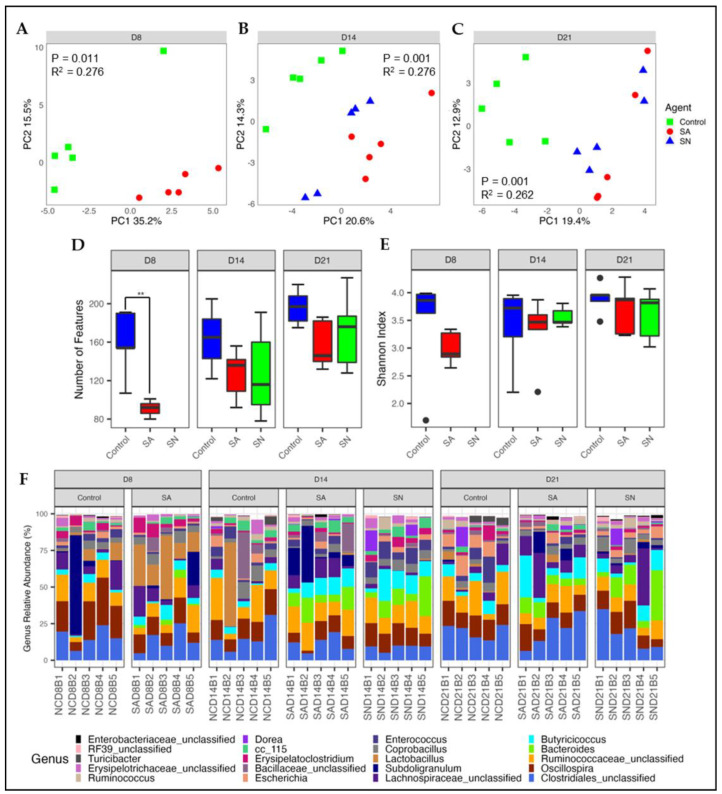
Effect of *Salmonella* colonization on bacterial diversity. PCA plots were used to visualize differences in bacterial composition between control, *Salmonella* artificial (SA), and *Salmonella* natural (SN) groups on days 8 (**A**), 14 (**B**), and 21(**C**), as measured by the Euclidean distance between samples. Changes in community richness and evenness were determined using the Observed Features (**D**) and Shannon diversity (**E**) calculators, respectively. Relative abundance was calculated using CSS normalized data, and the top 20 genera are shown (**F**). Significant differences were determined using the permanova and Kruskal-Wallis tests for beta and alpha diversity, respectively. Wilcoxon Rank-Sum tests were used for pairwise comparisons in richness and evenness. Results were considered significant at a *p* < 0.05. Samples falling outside of the first and third quartiles are shown as black dots.

**Figure 5 animals-12-02867-f005:**
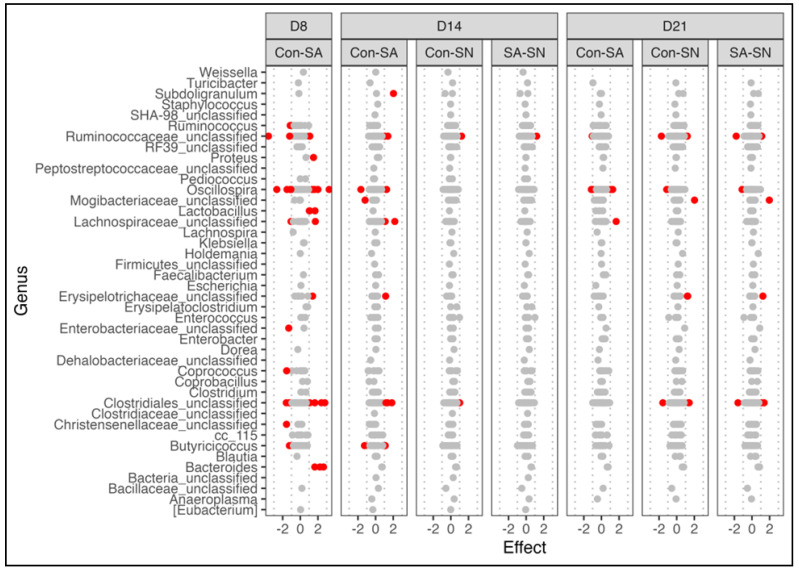
Shifts in the abundance of bacterial features in response to *Salmonella* colonization. At 7 days of age, control and SA birds were given an oral gavage of PBS or ST, respectively. Following inoculation, five SA birds were moved to pens with uninoculated birds (SN) to act as *Salmonella* seeder birds for natural horizontal transmission. Changes in the abundance of ASVs were determined using ALDEx2. Genera with features significantly affected in at least one comparison were plotted. Each dot represents the calculated effect size of a single ASV with significantly affected ASVs colored red. Differences were considered significant at FDR < 0.05 and effect > 1.

**Figure 6 animals-12-02867-f006:**
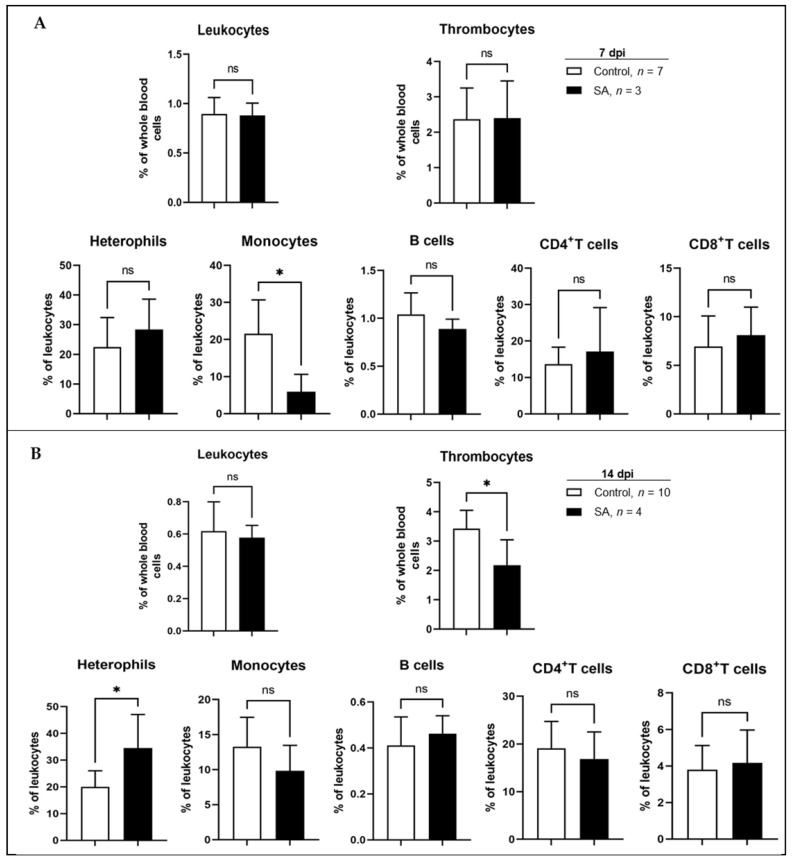
Effect of *Salmonella* inoculation on leukocyte populations. Whole blood, collected from 10 birds at 7 (**A**) and 14 (**B**) days post-infection, was subjected to flow cytometry to identify changes in leukocyte subpopulations. Samples displaying high amounts of clotting prior to processing were excluded. Significant differences were determined using the Student’s *t*-test and results were considered significantly different at *p* < 0.05 (ns = Not Significant, * = *p* < 0.05).

## Data Availability

Sequencing data in this study are openly available in NCBI SRA at BioProject number PRJNA875154. Additional data are available upon request from the corresponding author.

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
