# Peer review of "Effect of Salmonella Typhimurium Colonization on Microbiota Maturation and Blood Leukocyte Populations in Broiler Chickens"

_animals, 2022, doi:10.3390/ani12202867_

Round 1

Reviewer 1 Report

Thank you for giving me an opportunity to review Effect of Salmonella Typhimurium colonization on microbiome maturation and blood leukocyte populations in broiler chicken. The paper is interesting in few aspects but requires some modification before publications. 

The abstract is terribly needs major modifications. It is more general and not specific. The authors need to reorganize the abstact again but giving detailed results and materials and methods and clear cut conclusion based on the results. 

environmetnal temperature and other factors must be described in materials and methods.

discussion seems to be inappropriate and small. 

Again conclusion is very general. 

Author Response

Thank you for giving me an opportunity to review Effect of Salmonella Typhimurium colonization on microbiome maturation and blood leukocyte populations in broiler chicken. The paper is interesting in few aspects but requires some modification before publications. 

Comment 1: The abstract is terribly needs major modifications. It is more general and not specific. The authors need to reorganize the abstract again but giving detailed results and materials and methods and clear cut conclusion based on the results. 

Response 1: Thank you for the feedback, the abstract has been reworded as suggested.

Comment 2: Environmental temperature and other factors must be described in materials and methods.

Response 2: We have included the details in lines 127 to 129 with the appropriate reference.

Comment 3: Discussion seems to be inappropriate and small. 

Response 3: Additional information has been added to the discussion as requested.

Comment 4: Again conclusion is very general. 

Response 4: The authors recognize the general nature of the conclusions. However, we feel the conclusions provided are appropriate for the goal of this study which was to provide a survey of the effect of Salmonella Typhimurium on the intestinal microbiota.

Reviewer 2 Report

Thanks for this intersting paper. Please find small comments below :

Missing auhors and affiliation on first page

Figure 2, 3, 4 : all histograms do not sum to 100%.

Figure 4 : please define SN, SA in the caption of the figure

Figure 6 : write the day of of sampling direclty on the figure / number of replicates for « SA » group is very small

Author Response

Thanks for this interesting paper. Please find small comments below :

Comment 1: Missing authors and affiliation on first page

Response 1: Thank you for bringing the error to our attention. We have ensured that the authors and affiliations are present.

Comment 2: Figure 2, 3, 4 : all histograms do not sum to 100%.

Response 2: It is correct that the histograms in figures 2 and 4 do not sum to 100%. This is because only the top 20 genera are shown. We decided against combining the remaining genera into an “Others” category as the top 20 account for more than 95% of the data in the majority of samples which would make visualization of the “Others” category difficult.

Figure 3 has been corrected to show all histograms at 100%.

Comment 3: Figure 4 : please define SN, SA in the caption of the figure

Response 3: The figure legend has been updated as suggested.

Comment 4: Figure 6 : write the day of of sampling direclty on the figure / number of replicates for « SA » group is very small

Response 4: The day of sampling has been added to the figure legend in each pan and the number of replicates text size has been increased.

Reviewer 3 Report

This is an interesting study looking at gut microbiota composition and leukocyte populations in blood, in the context of Salmonella Typhimurium infection

I am a little unsure about your use of the word “microbiome” in the manuscript as you have essentially looked at microbiota, specifically bacteria, composition.

Line 27 – “…in colonisation, ..” Is the suggestion they facilitate colonisation?

Lines 57-60 – Some more discussion here could be helpful? For example, some suggestions of (viable?) microbial (or components) transfer pre-hatch vs inadvertent contamination.

Line 62 – nest environments?

Line 64 – Could you define “Once established,..”?

Materials & methods

Do you have any information about the Salmonella status of the hens/parents?

Mixed sex? History (e.g. any vaccinations, etc.)?

Line 195 – Do you mean two of the three groups of birds investigated?

Line 200 – “two of the three samples,” It is a little unclear. I understood “Samples were collected from five birds per treatment at each time point”.

Line 210 – “..using 3 to 5 birds per age” Were samples/birds lost? Has this been explained?

Line 219 – “A numerical increase…” This is not absolute and should be clarified/revised?

Line 310-312 – Did you report total numbers in section “3.1.3. Alteration in proportions among leukocyte populations with age”?

Line 321-322 – Can you clarify numbers of samples excluded?

With leukocyte populations, the data provided are proportions or percentage. Some of the language used is a bit confusing at times as it implies (absolute) numbers changed but we do not have these data (presented)?

Lines 356-358 – This perhaps also suggests that the infection dynamics are (unsurprisingly) different between natural and artificial colonisation, with the former being less dramatic?

Could you suggest ways in which this “information that can be used to develop effective methods or antibiotic alternatives to control or reduce Salmonella in chickens”?

Author Response

This is an interesting study looking at gut microbiota composition and leukocyte populations in blood, in the context of Salmonella Typhimurium infection

Comment 1: I am a little unsure about your use of the word “microbiome” in the manuscript as you have essentially looked at microbiota, specifically bacteria, composition.

Response 1: The reviewer brings up a good point. The word “microbiome” has been changed to “microbiota” throughout the manuscript.

Comment 2: Line 27 – “…in colonisation, ..” Is the suggestion they facilitate colonisation?

Response 2: The sentence is meant to suggest the host immune system’s attempt to fight colonization and has therefore been changed from “in colonization’ to “during colonization”. (Line 51)

Comment 3: Lines 57-60 – Some more discussion here could be helpful? For example, some suggestions of (viable?) microbial (or components) transfer pre-hatch vs inadvertent contamination.

Response 3: Further information on evidence of microbial transfer pre-hatch has been added in lines 83 – 87.

Comment 4: Line 62 – nest environments?

Response 4: The spelling error has been corrected. (Line 92)

Comment 5: Line 64 – Could you define “Once established,..”?

Response 5: The intestinal microbiota has been shown to mature and stabilize with age as the resident bacteria fill their respective niches. For clarity, the sentence has been changed to “Following stabilization of microbiota composition,…” (Line 94)

Materials & methods

Comment 6: Do you have any information about the Salmonella status of the hens/parents?

Response 6: Unfortunately, there was no way to determine the Salmonella status of the parental flock for these birds. However, the control birds were determined to be Salmonella negative up to 21 days of age which would indicate that no vertical transmission occurred in these birds.

Comment 7: Mixed sex? History (e.g. any vaccinations, etc.)?

Response 7: The sex and vaccination history of the chicks was added on lines 122-125.

Comment 8: Line 195 – Do you mean two of the three groups of birds investigated?

Response 8: Two of the three birds investigated is correct as the sentence is only referring to the day-of-hatch samples. The sentence has been updated to, “the cecal microbiota at day-of-hatch was dominated by…” (Line 231)

Comment 9: Line 200 – “two of the three samples,” It is a little unclear. I understood “Samples were collected from five birds per treatment at each time point”.

Response 9: Thank you for pointing out the confusion. Five birds per treatment were sampled on all days, however, only 2 samples from day 0 were discarded due to poor quality. The methods section has been updated for clarity (Line 137-138).

Comment 10: Line 210 – “..using 3 to 5 birds per age” Were samples/birds lost? Has this been explained?

Response 10: Two samples were discarded due to unacceptable quality. This has been clarified in the materials and methods section as well as in the results (Line 232)

Comment 11: Line 219 – “A numerical increase…” This is not absolute and should be clarified/revised?

Response 11: The sentence has been revised to “A numerical increase in the percentage of…”

Comment 12: Line 310-312 – Did you report total numbers in section “3.1.3. Alteration in proportions among leukocyte populations with age”?

Response 12: The numbers reported were changes in percentage of leukocytes. The text has been updated for clarity.

Comment 13: Line 321-322 – Can you clarify numbers of samples excluded?

Response 13: As the flow cytometry protocol in this experiment depended upon whole blood, samples displaying high amounts of clotting were discarded from the analysis. This has been further clarified on lines 349-351.

Comment 14: With leukocyte populations, the data provided are proportions or percentage. Some of the language used is a bit confusing at times as it implies (absolute) numbers changed but we do not have these data (presented)?

Response 14: The data provided are percentages. The word proportion has been changed to percentage for clarification.

Comment 15: Lines 356-358 – This perhaps also suggests that the infection dynamics are (unsurprisingly) different between natural and artificial colonisation, with the former being less dramatic?

Response 15: Yes, this is correct. The text has been updated to bring greater attention to the dynamics (Line 401-402).

Comment 16: Could you suggest ways in which this “information that can be used to develop effective methods or antibiotic alternatives to control or reduce Salmonella in chickens”?

Response 16: The relationships seen here between Salmonella and E. coli and Bacteroides are of particular interest and warrant further investigation into the potential for E. coli and Bacteroides to serve as probiotics against Salmonella. The original statement in the simple summary has been revised and further information on this has been added to the discussion.

Reviewer 4 Report

The study investigated the effect of Salmonella Typhimurium colonization on microbiome maturation and blood leukocyte populations in broiler chickens. Comments are below.

Overall

1.        The sample number is not enough. And the age of chicken is too young, and the market-aged birds should be used as infection from meat is one of the routes.   

2.        Age-dependent cecal microbiota changes were measured, but the chicken for the analysis was different between the analysis day. A fecal sample should therefore be used because it does not need to sacrifice, and it can analyze the microbial changes in one living chicken.

Simple Summary

1.        Line 10-11: The relationship is unclear, so insert “,” after Salmonella.

Introduction

2.        1st paragraph: Make it compact.

3.        State the reason of leucocyte measurements more in terms of Salmonella control for preventing the infection in humans.  

Discussion

4.        It should discuss the difference between the current results and other studies using Salmonella infection.

Author Response

The study investigated the effect of Salmonella Typhimurium colonization on microbiome maturation and blood leukocyte populations in broiler chickens. Comments are below.

Overall:

Comment 1: The sample number is not enough. And the age of chicken is too young, and the market-aged birds should be used as infection from meat is one of the routes.   

Response 1: While increased sampling is always better, when possible, the authors believe the current number is sufficient to convey the changes in microbiota composition and dynamics.

As for the age of the chickens, the goal of this study was to understand changes in the microbiota during colonization, not at slaughter. As multiple studies have demonstrated the dynamic nature of the intestinal microbiota (Gao et al., 2-17, Oakley and Kogut, 2016, Proszkowiec-Weglarz et al., 2022), we believe that sampling during the first two weeks after colonization is sufficient to address our research goals for this experiment.

Gao, P., Ma, C., Sun, Z., Wang, L., Huang, S., Su, X., Xu, J. and Zhang, H., 2017. Feed-additive probiotics accelerate yet antibiotics delay intestinal microbiota maturation in broiler chicken. Microbiome5(1), pp.1-14.

Oakley, B.B. and Kogut, M.H., 2016. Spatial and temporal changes in the broiler chicken cecal and fecal microbiomes and correlations of bacterial taxa with cytokine gene expression. Frontiers in veterinary science3, p.11.

Proszkowiec-Weglarz, M., Miska, K.B., Ellestad, L.E., Schreier, L.L., Kahl, S., Darwish, N., Campos, P. and Shao, J., 2022. Delayed access to feed early post-hatch affects the development and maturation of gastrointestinal tract microbiota in broiler chickens. BMC microbiology22(1), pp.1-20.

Comment 2: Age-dependent cecal microbiota changes were measured, but the chicken for the analysis was different between the analysis day. A fecal sample should therefore be used because it does not need to sacrifice, and it can analyze the microbial changes in one living chicken.

Response 2: The reviewer is correct that fecal samples allow for repetitive sampling of the same chicken, studies comparing microbiota composition from cecal and fecal (cloacal swab) samples from the same bird have revealed an inability of fecal samples to accurately represent the quantitative changes in the cecum (Stanley et al., 2015, Williams and Athrey, 2020), particularly as it pertains to more lowly abundant taxa (Andreani et al., 2020). Therefore, euthanizing and sampling the ceca of different birds at each time point is still the most accurate way to measure changes in the cecal microbiota.

Andreani, N.A., Donaldson, C.J. and Goddard, M., 2020. A reasonable correlation between cloacal and cecal microbiomes in broiler chickens. Poultry Science, 99(11), pp.6062-6070.

Stanley, D., Geier, M.S., Chen, H., Hughes, R.J. and Moore, R.J., 2015. Comparison of fecal and cecal microbiotas reveals qualitative similarities but quantitative differences. BMC microbiology, 15(1), pp.1-11.

Williams, T. and Athrey, G., 2020. Cloacal swabs are unreliable sources for estimating lower gastro-intestinal tract microbiota membership and structure in broiler chickens. Microorganisms, 8(5), p.718.

Simple Summary

Comment 3: Line 10-11: The relationship is unclear, so insert “,” after Salmonella.

Response 3: The sentence has been changed as suggested

Introduction

Comment 4: 1stparagraph: Make it compact.

Response 4: The paragraph length has been reduced.

Comment 5: State the reason of leucocyte measurements more in terms of Salmonella control for preventing the infection in humans.  

Response 5: The section has been updated as suggested on lines 107-108

Discussion

Comment 6: It should discuss the difference between the current results and other studies using Salmonella infection.

Response 6: Additional information on the similarities and differences between our results and the published literature on Salmonella infection has been added.